



# Hemispheric asymmetries in recent changes of the stratospheric circulation

Felix Ploeger[1,2] and Hella Garny[3,4]

[1]Institute for Energy and Climate Research: Stratosphere (IEK–7), Forschungszentrum Jülich, Jülich, Germany.
[2]Institute for Atmospheric and Environmental Research, University of Wuppertal, Wuppertal, Germany.
[3]Deutsches Zentrum für Luft- und Raumfahrt (DLR), Institut für Physik der Atmosphäre, Oberpfaffenhofen, Germany.
[4]Ludwig Maximilians University of Munich, Meteorological Institute Munich, Munich, Germany.

**Correspondence:** Felix Ploeger (f.ploeger@fz-juelich.de)

**Abstract.** Despite the expected opposite effects of ozone recovery, the stratospheric Brewer-Dobson circulation (BDC) has been found to weaken in the Northern hemisphere (NH) relative to the Southern hemisphere (SH) in recent decades, inducing substantial effects on chemical composition. We investigate hemispheric asymmetries in BDC changes since about 2000 in simulations with the transport model CLaMS driven with different reanalyses (ERA5, ERA-Interim, JRA–55, MERRA–2)

and contrast those to a suite of free-running climate model simulations. We find that age of air increases robustly in the NH stratosphere relative to the SH in all reanalyses considered. Related nitrous oxide changes agree well between reanalysis-driven simulations and satellite measurements, providing observational evidence for the hemispheric asymmetry in BDC changes. Residual circulation metrics further show that the composition changes are caused by structural BDC changes related to an upward shift and strengthening of the deep BDC branch, resulting in longer transit times, and a downward shift and weakening

shallow branch in the NH relative to the SH. All reanalyses agree on this mechanism. Although climate model simulations show that ozone recovery will lead to overall reduced circulation and age of air trends, the hemispherically asymmetric signal in circulation trends is small compared to internal variability. Therefore, the observed circulation trends over the recent past are not in contradiction to expectations from climate models. Furthermore, the hemispheric asymmetry in BDC trends imprints on the composition of the lower stratosphere and the signal might propagate into the troposphere, potentially affecting composition

down to the surface.

## 1 Introduction

The Brewer-Dobson circulation (BDC) is the global transport circulation in the stratosphere (Brewer, 1949) and is a key control factor for the chemical composition of that region, and even down to the surface (Ray et al., 2019). The BDC is characterized by large-scale upwelling in the tropical lower stratosphere, subsequent poleward transport and downwelling at middle and

high latitudes (Holton et al., 1995; Butchart, 2014). Climatological patterns as well as trends in the BDC are tightly coupled to patterns and trends in atmospheric waves on different scales, which induce the circulation's driving force when breaking (Haynes et al., 1991). These waves induce advection in the meridional latitude-altitude plane and also quasi-horizontal eddy mixing along isentropic surfaces. Disentangling the effects of advection by the residual circulation and mixing is important for





interpreting the BDC's effect on chemical composition (e.g., Garny et al., 2014), and when comparing different models (e.g.,
Eichinger et al., 2019).

Climate models robustly predict a strengthening and acceleration of the BDC in a warming future climate (e.g., Abalos et al., 2021). This circulation strengthening manifests in a global decrease of mean age of air (AoA), the average transit time through the stratosphere (Waugh and Hall, 2002). In the models, the BDC strengthening is largely caused by a strengthening and upward shift of the subtropical jets in a changing climate and a related upward shift of the critical layers for wave breaking (Garcia and
Randel, 2008; McLandress and Shepherd, 2009; Shepherd and McLandress, 2011). Although the BDC strengthening is one of the more robust aspects of climate model predictions (Butchart, 2014), clear evidence from observations is still lacking. In particular, AoA deduced from balloon-borne tracer observations in the middle stratosphere shows insignificant trends over the last decades (Engel et al., 2009; Fritsch et al., 2020).

In a recent paper, Polvani et al. (2018) have shown from climate model simulations that changes in ozone-depleting sub-
stances (ODSs), like chlorofluorocarbons (CFCs), substantially affect trends in the BDC. These ODS-induced BDC changes are rather related to the ODS-induced changes in ozone than to direct radiative effects (Abalos et al., 2019). During the period of ozone depletion (before about the year 2000), when ODS concentrations were increasing due to anthropogenic emissions, the ODS increase caused a strengthening of the BDC and hence an intensification of the BDC's response to climate change. During the ozone recovery period after about the year 2000, on the other hand, the ODS decline caused a BDC weakening and
hence an attenuation of its climate change response (Polvani et al., 2018). Because polar ozone depletion is strongest in the SH, also the ODS effect on the BDC is stronger in the SH than NH, and the ODS effects cause hemispheric asymmetries in BDC trends. As during the ozone depletion period (before about 2000) the ODS increase amplifies the GHG-induced strengthening of the BDC, with this strengthening being largest in the SH, the NH circulation weakens relative to the SH (or equivalently, the SH circulation strengthens relative to the NH). During ozone recovery (post 2000), the ODS effect counteracts the GHG-
induced BDC strengthening, maximum in the SH, and climate models predict a strengthening NH circulation relative to the SH. This ODS-induced hemispheric asymmetry of BDC trends, with a weakening circulation in the NH before 2000 and a strengthening BDC after 2000 relative to the SH, appears to be a robust prediction from climate models when simulating the long-term evolution of the climate (Polvani et al., 2018).

The observational record of the ozone recovery period extends by now to about 20 years. Over periods of only one to two
decades, it is a particular challenge to understand and attribute trends in circulation and transport. Medium-term variations in the BDC may be related to a complex interplay between different modes of natural variability, such as El Niño Southern Oscillation, ENSO (Diallo et al., 2019), the stratospheric Quasi-Biennial Oscillation, QBO (Ray et al., 2019), variations in stratospheric aerosol (Diallo et al., 2017), and the Pacific Decadal Oscillation (Iglesias-Suarez et al., 2021). For studying decadal trends, meteorological reanalysis are of particular value as they represent the observed atmospheric evolution and
provide a complete and consistent data set. Nevertheless, even the newest generation of reanalyses has been found to be subject to large uncertainty concerning decadal BDC trends, with different reanalyses even showing opposite changes during certain periods (Chabrillat et al., 2018; Ploeger et al., 2019).



Atmospheric trace gas observations indicate a BDC weakening and slow-down in the NH relative to the SH since about the year 2000. This hemispheric asymmetry in BDC trends post-2000 can be seen in increasing HCl mixing ratios (Mahieu et al., 2014), decreasing nitrous oxide (Nedoluha et al., 2015; Han et al., 2019), increasing stratospheric fluorine (Prignon et al., 2021), as well as increasing total columns for HCl and $HNO_3$ (Strahan et al., 2020) in the NH relative to the SH. The trace gas changes appear related to a hemispheric asymmetry (difference) in mean age changes deduced from satellite observations (Stiller et al., 2017), with mean age increasing in the NH and decreasing in the SH lower stratosphere. This recent BDC increase in the NH relative to the SH is in contrast to expectations from the forced signal by ozone recovery. It is currently an open question whether this apparent discrepancy is an actual contradiction to climate model projections, or whether recent BDC changes might simply be a result of natural variability.

In this paper, we investigate the hemispheric asymmetry in BDC trends since about the year 2000. In particular, we contrast the climate model-based expectation of a strengthening NH circulation relative to the SH with the trends found in different reanalyses and in satellite trace gas observations. Comparing four different reanalyses and a suite of climate model simulations enables us to assess the robustness of the representation of BDC changes. Based on different circulation diagnostics (residual circulation, age of air) we also analyze the related dynamical factors. Specific research questions for this paper are:

(i) How robust are hemispheric asymmetries (differences) in BDC trends over the last about two decades?

(ii) Which processes cause these hemispheric asymmetries?

(iii) Are the observed recent hemispherically asymmetric BDC changes in contradiction to expectations from ozone recovery as simulated by climate models?

The used data sets, models and methods are described in Sect. 2. Thereafter, Section 3 presents AoA trends and their hemispheric asymmetry based on 4 different reanalysis products and compares the related nitrous oxide trends to satellite observations. In Sect. 4 we investigate the underlying changes in circulation and dynamics. The results from reanalyses and observations are compared with results from climate models in Sect. 5 and are further discussed in Sect. 6. The main conclusions are summarized in Sect. 7 to answer the three specific research questions raised above.

## 2  Data and method

### 2.1  Reanalysis data sets and CLaMS transport model

For the investigation of BDC changes in reanalyses we apply the Chemical Lagrangian Model of the stratosphere CLaMS (McKenna et al., 2002, e.g.,). As a Lagrangian offline chemistry transport model, CLaMS is based on 3D air parcel trajectories, which are calculated in a hybrid vertical coordinate system, being purely diabatic in the stratosphere (potential temperature as vertical coordinate) and transforming into an orography-following $\sigma$-coordinate below (for details see Pommrich et al., 2014). In addition to advection, CLaMS includes a parameterization for small-scale atmospheric mixing processes depending on the





shear in the large-scale flow, such that in regions of large flow deformations strong mixing occurs. Horizontal winds and total diabatic heating rates for driving vertical transport are taken from reanalysis data (e.g., Ploeger et al., 2021).

BDC changes are investigated in four different reanalysis data sets, including the two most recent reanalyses from the European Centre for Medium-Range Weather Forecasts ECMWF, ERA5 (Hersbach et al., 2020) and ERA–Interim (Dee et al., 2011), the Japanese Meteorological Agency's reanalysis JRA–55 (Kobayashi et al., 2015), and the National Aeronautics and Space Administration's (NASA reanalysis MERRA–2 (Gelaro et al., 2017). Recent inter-comparisons between these reanalyses have been carried out within the Stratosphere-troposphere Processes And their Role in Climate (SPARC) Reanalysis Intercom-

parison Project S–RIP (Fujiwara et al., 2017). Although the general characteristics and variability of BDC transport are reliably represented in all these reanalysis products, recent comparisons have shown clear differences between reanalyses regarding the mean circulation strength and trends on decadal time scales (Chabrillat et al., 2018; Ploeger et al., 2019). Overall, JRA–55 has the fastest BDC and youngest mean age while MERRA–2 has the slowest BDC and oldest mean age, and ERA–Interim shows intermediate values. In particular, the newest reanalysis ERA5 has recently been shown to feature a slower BDC than

ERA–Interim (but faster than MERRA–2), somewhat slow-biased when compared to mean age in situ observations from NH middle latitudes (Ploeger et al., 2021).

    The CLaMS age of air simulations analysed here are the same as presented by Ploeger et al. (2021). The full stratospheric age spectrum is calculated using a pulse tracer method, with 60 tracer pulses released in the lowest model layer in the tropics (15°S-15°N) every other month (pulse length 30 days) and calculated over 10 years. From this tracer set the time-dependent

stratospheric age spectrum can be calculated at each model grid point and time step, with the spectra (transit time distributions) covering 10 years of transit time. As mean age calculated from these age spectra truncated at 10 years is low-biased, we deduce mean age from a clock-tracer which is linearly increasing at the surface (e.g., Hall and Plumb, 1994) and which includes an additional spin-up of 10 years by repeating the first available reanalysis year. As the analysis here focuses on the period after about 2000, the net spin-up for mean age is generally longer than 30 years.

The reanalysis-driven model simulations are compared to satellite observations in terms of $N_2O$ mixing ratios. The calculation of $N_2O$ in CLaMS includes chemical loss due to photolysis and reaction with $O(^1D)$, mainly occurring in the middle and upper stratosphere. Further details about the CLaMS chemistry scheme and model set-up are described by Pommrich et al. (2014).

    In addition to age of air, we use several residual circulation diagnostics throughout the paper. As a local measure for cir-

culation strength we consider residual circulation velocities $(\overline{v}^*, \overline{w}^*)$ and EP-flux divergence calculated from the reanalyses for estimating the wave driving (e.g., Butchart, 2014). As an integrated measure for the residual circulation we also calculated residual circulation transit times (RCTT) from 2D trajectories in the residual circulation (e.g., Birner and Bönisch, 2011; Garny et al., 2014). For consistency with the diabatic age of air calculation in CLaMS the residual circulation trajectories are calculated in isentropic coordinates using mass-weighted isentropic mean velocities (for further details, see Ploeger et al.,

2019). Moreover, the depth of the BDC is estimated from the minimum pressure $p_{min}$ encountered along residual circulation trajectories, such that a smaller $p_{min}$ value indicates a deeper circulation.



## 2.2 Climate models

To contrast the reanalysis-based findings on recent BDC changes with a suite of chemistry-climate model simulations, we use simulations provided through the Chemistry-Climate Model Initiative (CCMI) Phase 1 intercomparison project. Specifically, we use the REF–C2 simulations, that run from the past (1960) into the future (2100) under the RCP6.0 scenario (a detailed simulation description can be found in Eyring et al., 2013). The models employ either a coupled ocean model, or prescribe sea surface temperatures and sea ice concentrations from external climate model simulations. Thus, each simulation represents a different possible evolution of climate variability over the period of investigation in this study, and a suite of model simulations can therefore serve to quantify the role of internal variability for circulation changes. In this paper, we use a total of 9 REF–C2 simulations from 8 different models or model versions (CMAM, GEOSCCM, EMAC-L47MA, EMAC-L90MA, SOCOL3, MRI-ESM1r1, NIWA-UKCA, 2 ensemble member from ACCESS-CCM). For a detailed description of the individual models, we refer to Morgenstern et al. (2017).

## 2.3 Satellite observations

Nitrous oxide is a chemically long-lived species and can be used as a tracer for BDC transport (e.g., Minganti et al., 2020; Strahan et al., 2011). In this paper, we compare $N_2O$ mixing ratios simulated with CLaMS to measurements from Microwave Limb Sounder (MLS) on board the Aura satellite (Livesey et al., 2021) and from the Atmospheric Chemistry Experiment Fourier Transform Spectrometer (ACE–FTS) satellite instrument (Bernath et al., 2005). For both MLS and ACE–FTS data we analyze the measurement period 2005–2017.

MLS measurements have a comparatively high spatial sampling with about 3500 profiles per day (e.g., Livesey et al., 2021). Livesey et al. (2021) showed recently that a drift in MLS $N_2O$ has been significantly reduced in the newest version 5 data, which we use here. It was further argued that interregional changes, like the hemispheric asymmetry considered here, should not be affected by this drift. We carried out the analysis for this study also with the previous MLS data version 4 where the drift was more pronounced. As all our conclusions remained unchanged (not shown), the effects of the drift in MLS N2O are indeed negligible for the presented analysis. For the ACE–FTS instrument the spatial sampling is less dense (up to 15 sunrise and sunset solar occultations per day) compared to MLS, but the higher vertical resolution of 3-4 km (compared to 4-6 km for MLS) makes ACE–FTS $N_2O$ very valuable for studying transport in the lower stratosphere (e.g., Bernath et al., 2005; Strong et al., 2008).

For the analysis of this paper, monthly and zonal mean climatologies on pressure levels have been created from both satellite data sets and are compared to monthly, zonal mean climatologies from the CLaMS simulations. Trends are calculated from linear regression after deseasonalizing the monthly time series by subtracting the mean annual cycle at each latitude/level grid point. In the following, the hemispheric asymmetry (or difference) of trends of variable $X$ will be denoted $\Delta X$ (note that in some places the same notation indicates deseasonalized anomalies).

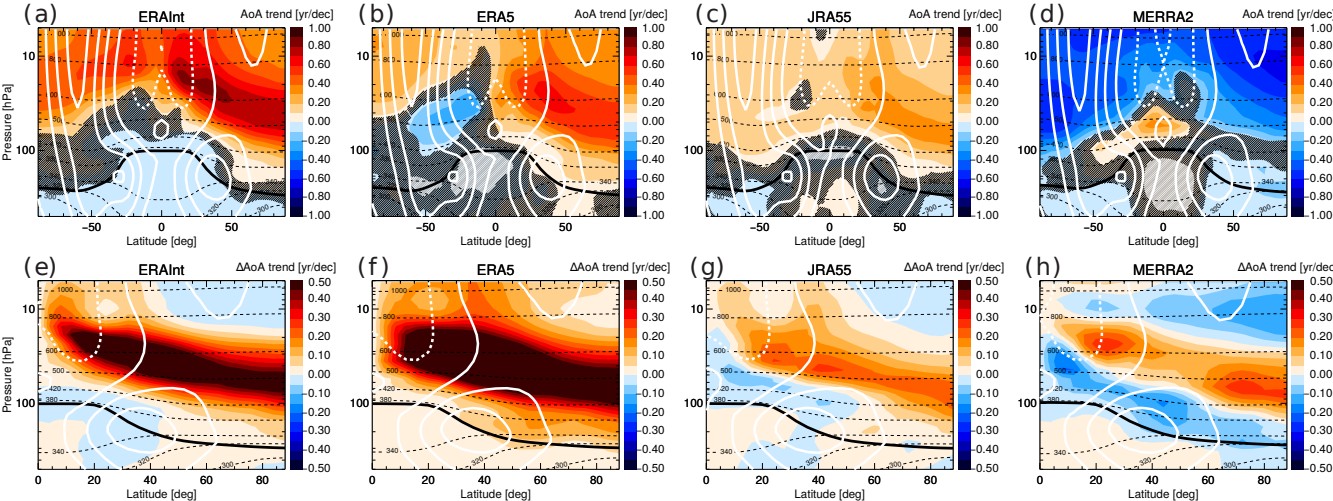

**Figure 1.** (a–d) Mean age trends for the period 2005–2017 for ERA5, ERA–Interim, JRA–55 and MERRA–2. (e–h) Hemispheric difference of mean age trends (NH minus SH) for different reanalyses, for the period 2005–2017. Grey lines show climatological age of air (a–d) and NH trend values (e–h, solid for positive, dashed for negative values), white lines show zonal wind contours (solid for positive, dashed for negative values, ±10 m/s steps), thin black dashed lines potential temperature levels and the thick black line the WMO tropopause.

## 3 Hemispheric asymmetries in reanalyses and observations

Figure 1 shows trends in stratospheric mean age of air calculated with CLaMS driven with different reanalyses for the period

2005–2017. The age trends in the two hemispheres are quite different for the different reanalyses. In the NH, ERA5, ERA–Interim and JRA–55 show positive trends (increasing age), but with different intensity, while MERRA–2 shows negative trends (decreasing age). In the SH, ERA5 and ERA–Interim show negative trends while JRA–55 shows positive trends, and MERRA–2 a layer of positive trends below a layer of negative trends. Although the trends in the respective hemispheres depend strongly on the choice of reanalysis, the hemispheric difference of trends (here NH minus SH) is remarkably robust (Fig. 1e–h). All

reanalyses show a large region of positive hemispheric difference in trends in the lower stratosphere between about 100 hPa and 10 hPa, sloping downward from equator to pole, where the NH age increases compared to the SH. The exact magnitude and regional extend of this trend differs somewhat between the data sets, but overall the hemispheric asymmetry of trends appears much more robust than the trends in the respective hemispheres.

The hemispheric asymmetry in mean age trends is indicative for differences in transport trends between the hemispheres and

is reflected in trends of long-lived trace gases. Figure 2 shows the trends in $N_2O$ together with the hemispheric asymmetry of trends from the four reanalysis-driven model simulations in comparison with MLS and ACE–FTS satellite measurements. As was evident already for mean age, also $N_2O$ trends are very different in the different reanalyses. ERA–Interim shows largely negative trends, stronger in the NH than SH, ERA5 and JRA–55 show negative trends in the NH and a diverse pattern in the SH (negative above positive trends in ERA5, insignificant trends in JRA–55), while MERRA–2 shows weakly negative trends in





**Figure 2.** (a–d) Nitrous oxide trends for the period 2005–2017 for ERA5, ERA–Interim, JRA55 and MERRA2, and from satellite measurements by the (e) MLS and (f) ACE–FTS instruments. (g–j) Hemispheric difference in nitrous oxide trends (NH minus SH) for the different reanalyses, and for (k) MLS and (l) ACE-FTS. Grey lines show climatological $N_2O$ (a–f) and the NH trend values (g–l, solid for positive, dashed for negative values), white lines show zonal wind contours (solid for positive, dashed for negative values), thin black dashed lines potential temperature levels and the thick black line the WMO tropopause.

the NH and weak positive trends in the SH. Trends in $N_2O$ are largely mirror images of the mean age trends (age increases are generally related to $N_2O$ decreases). This is related to the fact that $N_2O$ is a tropospheric tracer with sinks in the stratosphere such that a longer stratospheric transit time is related to stronger loss (e.g., Strahan et al., 2011). However, the relation between $N_2O$ and mean age trends is not unambiguous as mean age is more sensitive to the age spectrum tail than $N_2O$. For instance MERRA–2 shows weakly decreasing $N_2O$ coinciding with decreasing mean age in the NH.



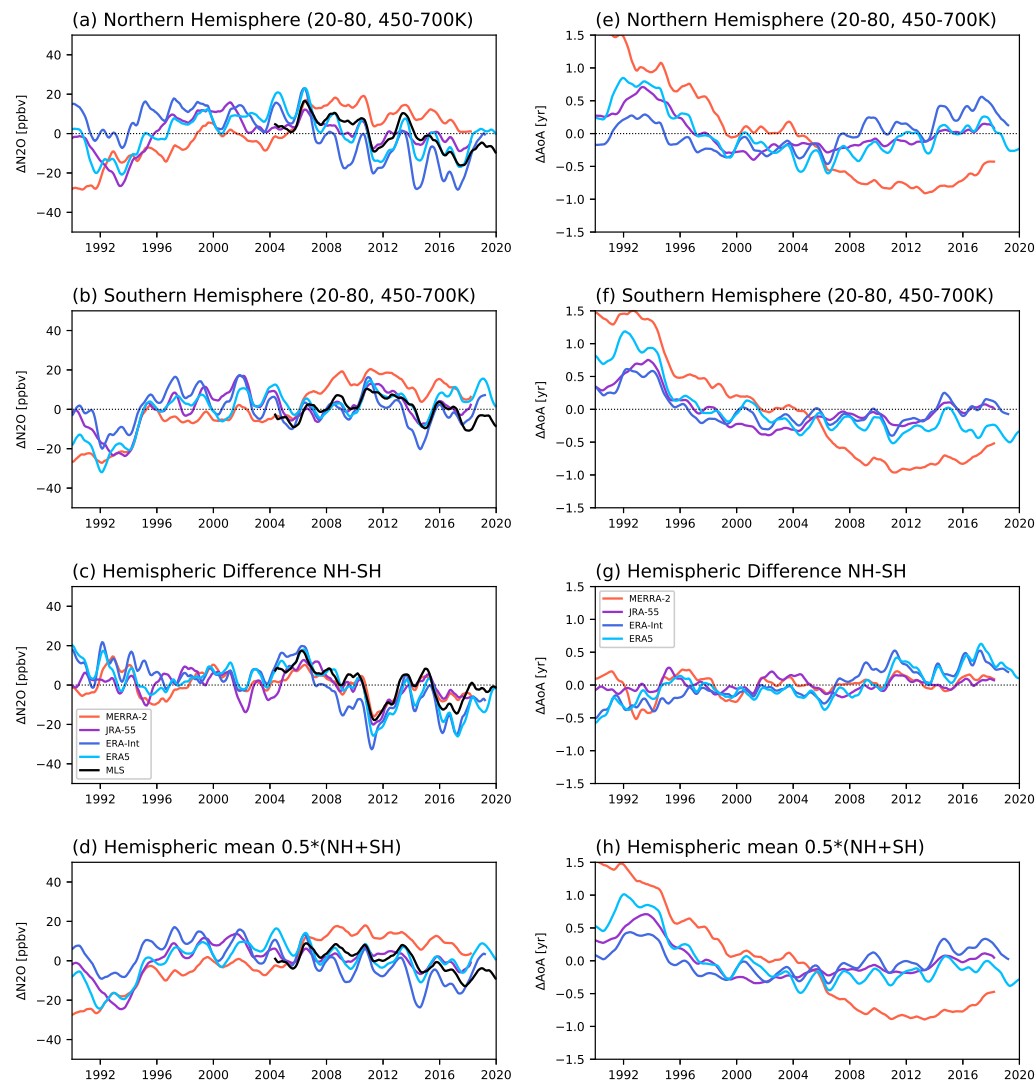

**Figure 3.** Time series for $N_2O$ in (a) NH, (b) SH, (c) the hemispheric difference (NH-SH), and (d) the hemispheric average (calculated as $0.5 \cdot (NH + SH)$), for the period 1990–2019, from CLaMS driven with different reanalyses and from MLS satellite observations. The data has been averaged zonally and over the region 450-700 K, 20–80°. (e–h) Same but for mean age from the CLaMS simulations.

The $N_2O$ changes derived from the two satellite data sets show differences in $N_2O$ trends of similar magnitude to the inter-reanalysis differences. In particular, in the SH N2O mixing ratios decrease for MLS while they increase for ACE-FTS data. The $N_2O$ decrease in the NH, however, clearly emerges from all model simulations and observations. As evident for mean age, also for $N_2O$ the hemispheric difference in trends from the four reanalysis-driven model simulations and measured by the MLS and ACE–FTS satellite instruments is much more robust than the trends in each hemisphere (Fig. 2). Clearly, the hemispheric





difference in $N_2O$ trends closely follows the mean age trends. The lower stratosphere region with positive hemispheric trend differences in age shows negative $N_2O$ hemispheric trend differences. In particular, there is good agreement in the trend signal between the reanalysis-driven model simulations and the two satellite instruments. For both $N_2O$ and age the signal in the hemispheric $\overline{v}^*$ trend difference appears to be dominated by changes in the NH. Quantitatively, ERA5 and ERA–Interim somewhat overestimate the trends while JRA–55 and MERRA–2 somewhat underestimate the trends when compared to the

measurements. However, given the good qualitative agreement in the large-scale patterns despite the large uncertainties in decadal BDC trends in each hemisphere (e.g., Chabrillat et al., 2018; Ploeger et al., 2019), we deem the agreement between the simulated and observed hemispheric asymmetry in trends to be a strong indication for the relative aging of the NH compared to the SH over the last two decades.

Time series of $N_2O$ in Fig. 3 in the NH and SH lower stratosphere (averaged between 20–80° latitude and 450–700 K poten-
tial temperature) show large inter-annual variability. A distinct oscillation with period of about two years indicates influence of the QBO. After about 2004, $N_2O$ mixing ratios in the NH start decreasing, whereas no clear trends are found in the SH, result-ing in a negative trend in the hemispheric difference (Fig. 3c). These lower values in $N_2O$ hemispheric differences during the last decade seem to be a persistent feature when compared to the preceding period. Comparison to the mean age hemispheric difference time series in Fig. 3e–g indicates a clear anti-correlation between $N_2O$ and age on different time scales, showing
that $N_2O$ mixing ratios are mainly responding to transport changes in this region. In particular, the persistent decrease in the $N_2O$ hemispheric difference coincides with a persistent increase in the mean age hemispheric difference during the last decade. Remarkable is the close alignment of $N_2O$ hemispheric difference time series (Fig. 3c) given the large differences between the different data sets in each hemisphere (Fig. 3a–b). Consequently, the symmetric contribution, calculated as the average of the time series in both hemispheres, includes most of the inter-reanalysis differences (Fig. 3d, h), as will be further discussed in
Sect. 6.

## 4 Changes in circulation and dynamics

### 4.1 Residual circulation

Related changes in the residual circulation are analysed in Fig. 4. This figure shows the trends in meridional velocity $\overline{v}^*$ and in Eliassen-Palm (EP) flux divergence during 2005–2017 in ERA5, together with the hemispheric difference of trends. Figure
4 shows only ERA5, but the trends in $\overline{v}^*$ and EP-flux are consistent among all four reanalyses and the related conclusions are robust. Here, $\overline{v}^*$ represents the residual circulation outflow out of the tropics and EP-flux divergence represents the wave drag-induced force of resolved waves to drive the BDC in the reanalysis. Negative EP-flux divergence represents a westward force in the zonal momentum balance, hence a deceleration of westerly flow, and forces residual circulation poleward motion (positive/negative $\overline{v}^*$ in NH/SH) in the meridional plane (e.g., Haynes et al., 1991).
The trends in Fig. 4a and c show a clear and consistent pattern, with negative EP-flux trends (increased wave driving) above about 10 hPa in the NH corresponding to positive $\overline{v}^*$ trends and thus increased poleward flow. Below about 10 hPa opposite trends occur. The patterns become even more clear and consistent for the hemispheric difference in trends (Fig. 4b,





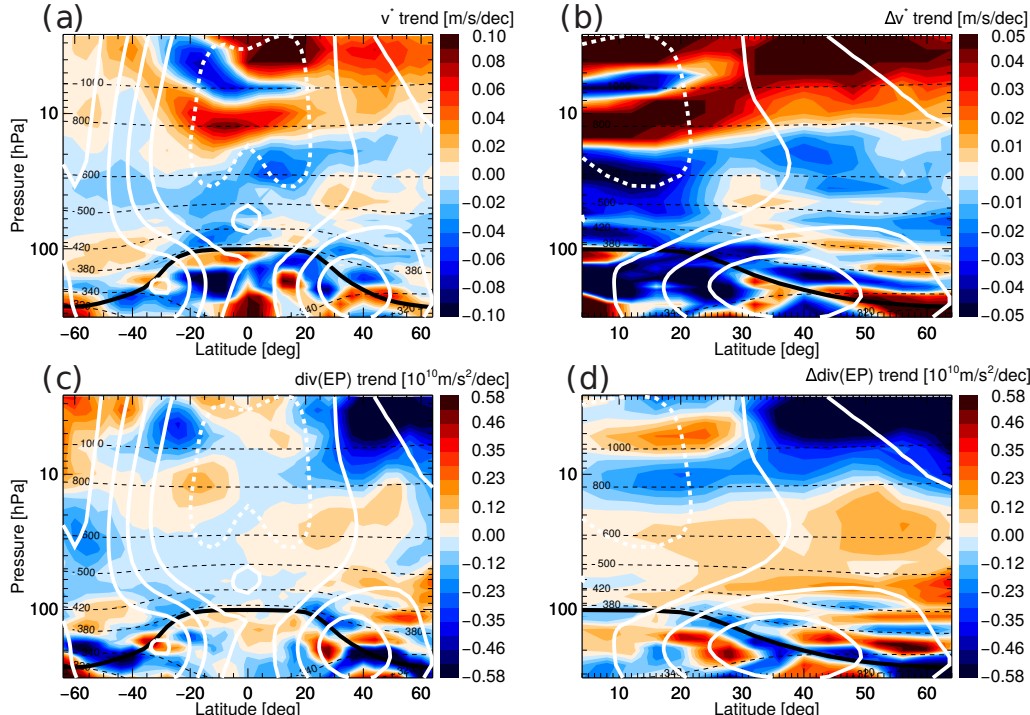

**Figure 4.** (a) Trend of meridional residual circulation velocity $\overline{v}^*$ for the period 2005–2017 from ERA5 reanalysis. (b) Hemispheric difference in $\overline{v}^*$ trend (NH minus SH). (c, d) Same but for Eliassen-Palm flux divergence. White lines show zonal wind contours (solid for positive, dashed for negative values), thin black dashed lines potential temperature levels and the thick black line the WMO tropopause.

d), showing increased wave driving (negative EP-flux trend) and residual circulation (positive $\overline{v}^*$ trend) above about 10 hPa in the NH compared to the SH, and opposite changes below. Moreover, seasonally resolved trends show that these patterns in the
hemispheric difference of trends are most pronounced in boreal winter season (December–February) in the NH (not shown).

Hence, residual circulation meridional velocity and EP-flux divergence changes are consistent, indicating a strengthening deep BDC branch above about 10 hPa and a weakening meridional circulation below, consistent with the trends deduced from age of air. It is remarkable how well even smaller scale features in the hemispheric difference (e.g., layered structures below about 70 hPa) in $\overline{v}^*$ and EP-flux divergence correlate in the reanalysis. In particular, comparison of the four different reanalyses
shows a very robust picture of the changes in wave driving (not shown).

Figure 5 presents indices for the strength of the deep and shallow circulation branches. The deep branch index is a measure of mid- to high latitude downwelling and is calculated as deseasonalized, annually averaged time series of vertical velocity $\overline{w}^*$ at 50–80° latitude and about 30 hPa ($\approx 25$ km). The shallow branch index is calculated similarly from horizontal outflow velocity $\overline{v}^*$ at turn-around latitudes between about 55–30 hPa ($\approx 20 - 25$ km). Similar metrics for the branches have been
recently applied by Han et al. (2019). The residual circulation velocity time series in the two hemispheres in Fig. 5a and b



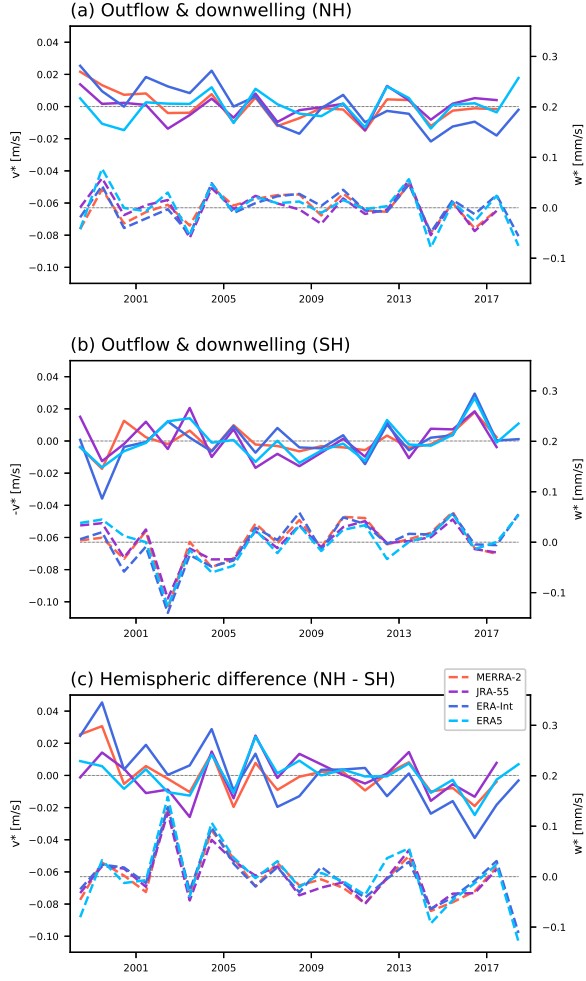

**Figure 5.** Indices for shallow and deep BDC branches for (a) NH, (b) SH, and (c) the hemispheric difference (NH–SH), calculated as deseasonalized, annually averaged time series of horizontal residual circulation outflow velocity $\overline{v}^*$ averaged over 20-25 km at turn-around latitudes (solid lines), and vertical velocity $\overline{w}^*$ averaged over 50–80° latitude at about 25 km (dashed). Different colours show different reanalysis data sets. Note that SH outflow velocities are shown with reversed sign ($-\overline{v}^*$) such that positive velocity corresponds to stronger outflow out of the tropics in both hemispheres.

show large year-to-year variability and no clear long-term change. However, the time series of hemispheric difference of these quantities, calculated as NH minus SH, show a negative trend over time for $\overline{w}^*$ indicating increasing deep branch downwelling, and a negative trend for $\overline{v}^*$ indicating weakening shallow branch outflow (Fig. 5c). Although the linear trends in $\overline{v}^*$ and $\overline{w}^*$ are not significant at 95% confidence level due to the large inter-annual variability and the short period considered (2005–2017), visual inspection of the time series in Fig. 5 and the agreement between the four different reanalyses suggests robust longer





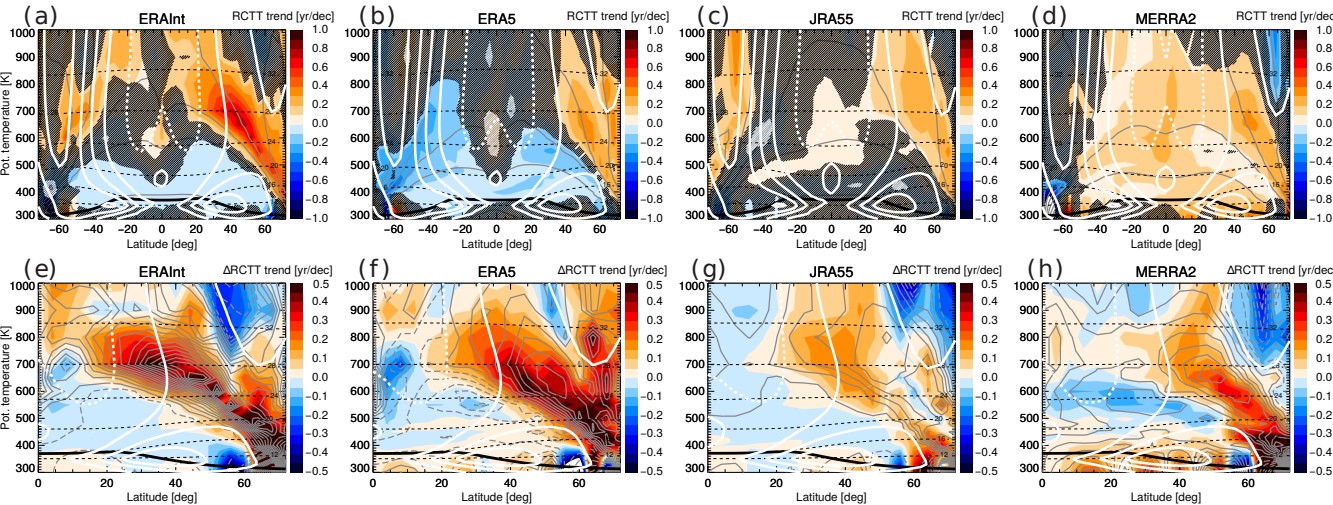

**Figure 6.** (a–d) Residual circulation transit time RCTT trends for the period 2005–2017 for ERA5, ERA–Interim, JRA55 and MERRA2. (e–h) Hemispheric difference in RCTT trends (NH minus SH) for the different reanalyses. Grey lines show climatological RCTT (a–d) and the NH trend values (e–h, solid for positive, dashed for negative values), white lines show zonal wind contours (solid for positive, dashed for negative values), thin black dashed lines log-pressure altitude levels and the thick black line the WMO tropopause.

term changes. Hence, the residual circulation velocities indicate that the deep branch strengthens over time since about 2000 in the NH relative to the SH, while the shallow branch weakens in NH relative to SH.

## 4.2 Structural circulation changes

The residual circulation represents the local (advective) circulation strength. However, transport times and tracer mixing ratios
depend on the spatial-temporal integral of this local circulation strength along an air parcel's path through the stratosphere. To analyse those integrated effects on transport times, we employ residual circulation transit times (RCTTs), the time scale for meridional transport along the residual circulation (see Sect. 2.1). Figure 6 shows the trends in RCTTs during 2005–2017. In the lower stratosphere, trends differ strongly between the different reanalyses, with ERA5 and ERA–Interim showing negative trends (accelerating circulation), MERRA–2 showing positive RCTT trends, and JRA–55 showing insignificant changes.
Above this region, on the other hand, a large area of positive RCTT trends robustly emerges in the NH stratosphere in all reanalyses. This pattern imprints on the hemispheric difference of trends, such that the most striking feature in the hemispheric difference is a region of positive RCTT trends above about 600 K (approximately 24 km) in the subtropics and sloping downward to about tropopause altitudes at high latitudes. The exact strength of these positive RCTT trends differs between the reanalyses, but qualitatively the signal is very robust. Below, in the region of the shallow BDC branch, RCTT trends are largely
negative, although the trend patterns in the respective hemispheres are not robust in the different reanalyses.

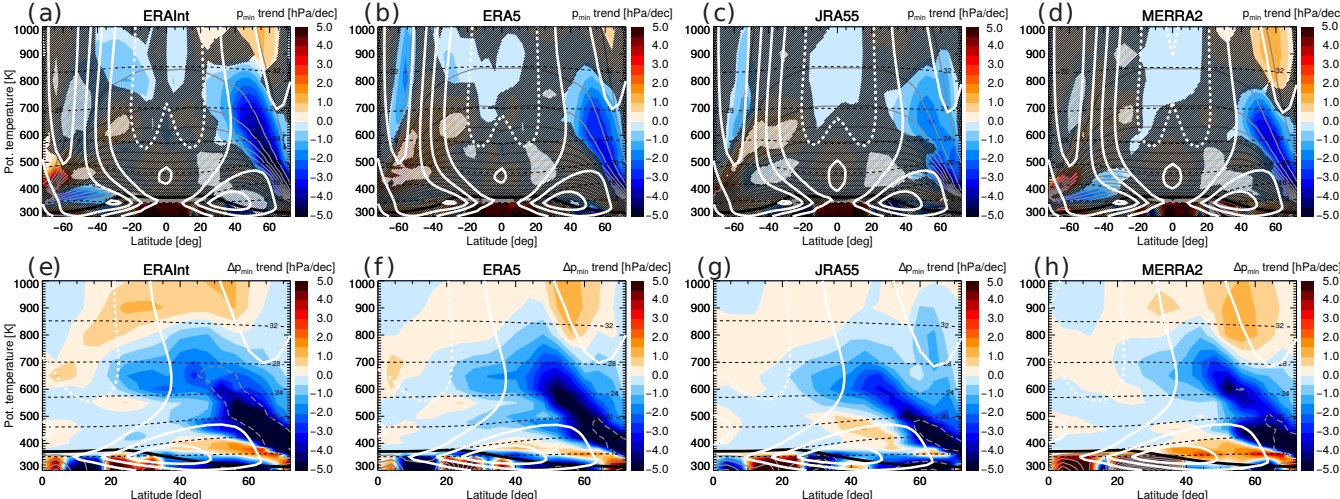

**Figure 7.** (a–d) Trends of minimium pressure along residual circulation trajectories $p_{min}$ for the period 2005–2017 for ERA5, ERA–Interim, JRA55 and MERRA2. (e–h) Hemispheric difference in $p_{min}$ trends (NH minus SH) for the different reanalyses. Grey lines show climatological $p_{min}$ (a–d) and the NH trend values (e–h, solid for positive, dashed for negative values), white lines show zonal wind contours (solid for positive, dashed for negative values), thin black dashed lines log-pressure altitude levels and the thick black line the WMO tropopause.

On first sight, the pattern of enhanced RCTTs in the NH relative to the SH might appear incompatible with the previous finding of a strengthened deep BDC branch with enhanced downwelling. However, as pointed out above, RCTTs are the temporally and spatially integrated effect of local circulation changes, and as such do not only depend on the local circulation strength, but also on changes in the pathways of air parcels, i.e. the structure of the BDC. To this end, Fig. 7e–h presents trends
in the minimum pressure $p_{min}$ along residual circulation trajectories. Birner and Bönisch (2011) proposed this diagnostic as a measure for the depth of the BDC and further argued that the deep branch affects latitudes poleward of about 60 degrees, whereas the shallow branch is restricted to lower latitudes. Comparison of Fig. 6 and Fig. 7 shows that the dominant signal of positive RCTT trends corresponds to negative $p_{min}$ trends in the NH, and in particular also in the hemispheric difference of NH changes relative to the SH. Decreasing minimum pressure along trajectories indicates that the circulation cell extends to higher
altitudes over time. Hence, the increase in RCTTs is related to an upward shift and strengthening of the deep BDC branch in the NH relative to the SH. Below, in the lowest stratosphere, changes are less robust between data sets, but still most reanalyses show a region of negative RCTT trends related to positive $p_{min}$ trends, likely indicating a downward shift and weakening of the shallow branch. In the lowest stratosphere, also changes in tropopause height could have additional effects on the RCTT trends.





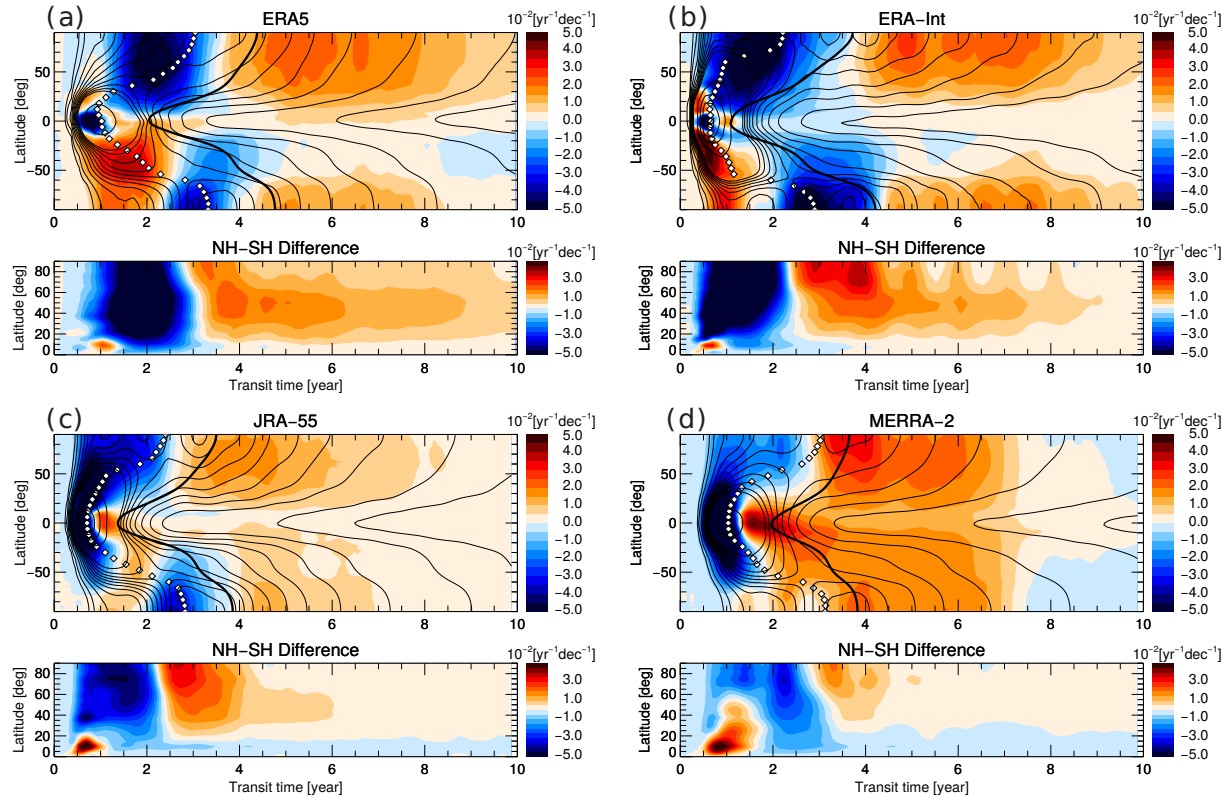

**Figure 8.** Age spectrum trends on the 500 K potential temperature surface over the period 2005–2017, for ERA5, ERA–Interim, JRA–55 and MERRA2. The bottom panels show the hemispheric difference of the trends (NH minus SH). Black thick lines show mean age, white symbols modal age, and the black thin lines the climatological age spectra.

## 4.3 Age of air spectra

In this section we consider age of air spectra to relate the structural changes in the circulation to the hemispherically asymmetric trend in mean age and chemical composition in the lower stratosphere. Figure 8 shows the age spectrum trend at 500 K potential temperature (about 20 km altitude) during 2005–2017, calculated with the CLaMS transport model driven by the four reanalyses. All reanalyses show a clear increase in the fraction of old air with transit times larger than about 3-4 years in both NH and SH. This change indicates a stronger contribution from the BDC deep branch, such that a larger amount of older air is found in the lower stratosphere over time. The trends for shorter transit times, indicative of the shallow BDC branch, show larger differences between the four data sets. In the NH, all reanalyses show a decrease in the fraction of young air, strongest for ERA5 and ERA–Interim, while even opposite changes occur in the SH in the different data sets.

The hemispheric difference of age spectrum trends shows a very robust pattern for all reanalyses (Fig. 8, lower panels). The fraction of air masses older than about 3 years (rather 2.5 years for ERA–Interim) increases more in the NH than in the SH,





while younger air masses show the opposite change. This trend towards a larger fraction of old air masses can be interpreted as a stronger contribution of air masses from the deep circulation branch and a weaker contribution of air masses with shallow transport pathways in the NH relative to the SH. Hence, age spectrum changes are consistent with the structural circulation changes inferred from other diagnostics and furthermore provide detailed information on involved time scales.

## 5  Climate model perspective

In this section the observation- and reanalysis-based findings regarding recent BDC trends are compared to climate model results. Free-running climate models are subject to a realization of natural variability which will be different from the observed variability. Hence, we use an ensemble of climate model simulations to investigate whether the hemispherically asymmetric signal in mean age trends, that is expected to occur in response to recovery from ODSs, can be expected to be detectable

compared to inter-annual variability.

Figure 9 shows the multi-model mean trends of mean age and their hemispheric difference from 9 different CCMI Ref-C2 model simulations for different periods. Consistent with previous work (Polvani et al., 2018), the mean age trends are generally stronger in the ozone depletion period (1965–2000) compared to the ozone recovery period (2000–2080). The trends over 1965–2000 also depict the hemispherically asymmetric signal with a stronger age decline in the SH compared to the

NH. While it can generally be expected that this signal reverses in the future, we find that the hemispheric asymmetry in the stratospheric age trends over the ozone recovery period (2000–2080) is very weak (Fig. 9h). This is further evident from the age trends averaged over the northern and southern hemisphere (70-10 hPa, 20-80°) as displayed in Fig. 10. The multi-model mean long-term mean age decrease in the past ozone depletion period is about twice as strong as the age trend in the ozone recovery period (2000–2080), in agreement with Polvani et al. (2018). The hemispheric difference in multi-model mean trends in the past

(1965–2000) is around 0.017 years/decade, and reverses to -0.005 years/decade for 2000–2080. This weaker asymmetric signal can simply be explained by the fact, that the ODS increase and associated ozone depletion occurred much more abrupt than the phase out of ODSs and ozone recovery will occur. Indeed, the total averaged hemispherically asymmetric mean age change over the ozone depletion period (i.e., the trend difference of $\sim 0.017\,\mathrm{yrs/decade} \cdot 3.6\,\mathrm{decades} = 0.06\,\mathrm{years}$) is similar to the total age change over the much longer recovery period (i.e., a trend difference of $\sim -0.005\,\mathrm{yrs/decade} \cdot 8.1\,\mathrm{decades} = 0.04\,\mathrm{years}$). Thus,

simply the fact that ozone recovery is slow compared to past ozone depletion explains that the hemispherically asymmetric signal in age trends per decade is considerably smaller in post-2000 periods.

For shorter time periods of two decades or less, internal climate variability enhances the spread of age trends between individual model simulations, as evident from Fig. 10. The multi-model mean age trend in a two-decade long ozone depletion period is slightly stronger compared to a period of same length from 2000–2020, and the hemispheric trend difference reverses

sign (see Fig. 10 and Fig. 9). However, individual models show considerable spread in the simulated trends and their hemispheric difference (see Fig. 10). In the period of 2005–2017, analyzed in terms of reanalysis and observational data in this study, multi-model mean trends are overall negative, but there is an even larger spread in the mean trends in either hemisphere,





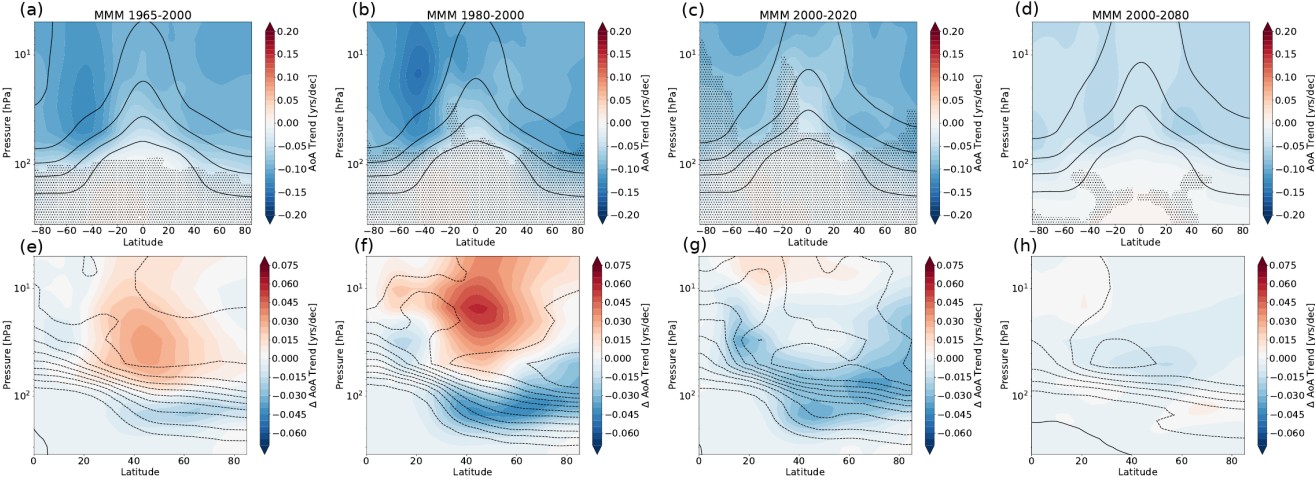

**Figure 9.** (a–d) Multi-model mean (MMM) trends in mean age from 9 CCMI REF–C2 model simulations for varying periods (see figure titles). (e–h) Hemispheric difference in the mean age trends. In the upper row, stippling indicates lack of significance of the trends (for the MMM trends, significance is detected if 2/3, i.e., 6 out of 9 model simulations show a significant trend).

ranging from about +0.1 to -0.3 yrs/decade. The hemispheric difference in trends spreads between positive and negative values, resulting in a multi-model mean trend difference close to zero.

Given the small hemispherically asymmetric signal in mean age trends in the ozone recovery period, the question arises when this signal can be expected to emerge from natural variability. To answer this question, Fig. 10 further shows periods of increasing length from 2000 to 2020, 2040 and 2080, respectively. In the 2000–2040 period, spread is reduced compared to 2000–2020, but the hemispheric difference in mean age trends still exhibits also positive values for individual model simulations. Even by 2080, 4 out of 9 model simulations display a hemispheric mean age trend difference of essentially zero. Thus,

based on the CCMI model ensemble, it is questionable when and whether a weaker BDC acceleration in the SH compared to the NH will be detectable from observational data.

## 6    Discussion

Different observations of a suite of trace gas species have suggested a slow-down of the BDC in the NH relative to the SH in the last decades since about the year 2000. These observations include measurements of HCl (Mahieu et al., 2014), $N_2O$

(Nedoluha et al., 2015), $SF_6$ for deducing mean age (Stiller et al., 2017) and $HNO_3$ (Strahan et al., 2020). Here we show that the underlying transport change is very robustly represented in current reanalyses data sets, including ERA5, ERA–Interim, JRA–55 and MERRA–2. Although transport changes, as estimated from mean age of air, may be fairly different between various reanalyses in each hemisphere, the hemispheric difference of trends is robust, showing a clear asymmetry with an aging NH relative to the SH.





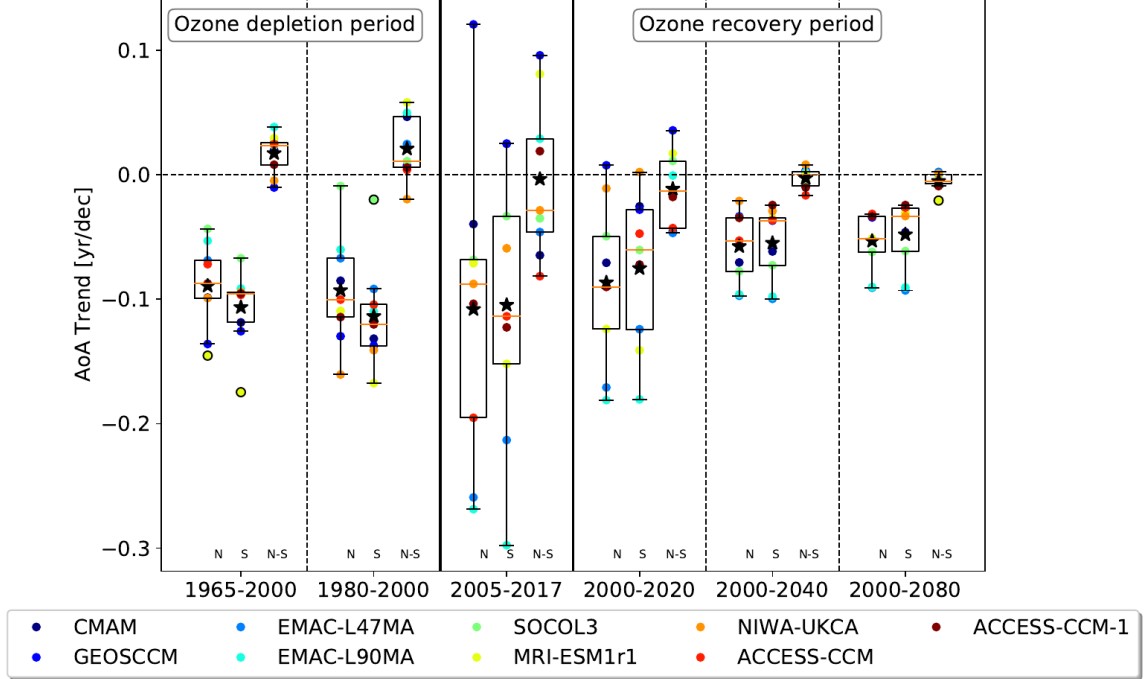

**Figure 10.** Mean age trends for different periods (see x-axis) averaged over 70-10 hPa and 20-80° of each hemisphere, with the left bar in each period displaying the NH trend, the middle bar the SH trend and the right bar the NH - SH difference. Black stars are the multi-model mean, and the boxes indicate the lower to upper quartiles with the median value indicated as orange line.

Han et al. (2019) proposed a strengthening deep BDC branch and a weakening shallow BDC branch in the NH relative to the SH as a cause for these observed trace gas changes. Our analysis of different transport diagnostics (e.g., age of air spectra, residual circulation velocities and trajectories) supports their result, and further shows that these circulation changes are robustly represented in the different reanalysis products. In particular, changes in the strength of deep branch extratropical downwelling show only little spread between reanalyses, while differences in shallow branch meridional outflow from the

tropics are somewhat larger (see Fig. 5). Analysis of residual circulation (2D) trajectories further shows that the BDC trends over the last two decades are characterized by a structural circulation change, rather than simply a variation in strength. Indeed, the deep circulation branch extends higher into the stratosphere and the shallow branch shifts to lower altitudes in the NH compared to the SH (Fig. 7). Therefore, the different reanalyses robustly show an upward shift of the deep branch and a downward shift of the shallow branch in the NH relative to the SH over the last two decades (since about 2000), related to

an upward shift of resolved wave drag. To what degree these changes are related to the southward circulation shift found in satellite measurements by Stiller et al. (2017) remains to be shown.

     From analysis of aircraft measurements, Bönisch et al. (2011) found an intensification of the shallow BDC branch after the year 2000. Our results are not in contradiction to this finding, as they concern relative changes between the hemispheres.



Moreover, Lin and Fu (2013) have defined the deep BDC branch being located above 30 hPa, the shallow BDC branch be-

tween 70–30 hPa and a transition branch between 100–70 hPa. Using this definition, the lowermost stratosphere considered by Bönisch et al. (2011) is largely affected by the transition branch. Regarding these lower layers, our analysis even shows a somewhat intensified resolved wave drag (Fig. 4) and an upward circulation shift (Fig. 7) in the NH relative to the SH, which is likely related to the intensified circulation found by Bönisch et al. (2011).

A main finding of this study is that hemispheric differences in BDC trends are much more robust than the absolute trends

in each hemisphere. Figure 3 shows that differences between reanalyses are strongly affecting the global mean stratospheric circulation whereas they cancel to some degree in the hemispheric difference. This finding seems plausible as such inter-reanalysis differences are likely related to differences and discontinuities in assimilated satellite data sets and have a global effect. MERRA–2 shows the most pronounced deviation from the other reanalyses in global mean age evolution, with strong and seemingly discontinuous decreases around 1995 and around 2005 Fig. 3). These discontinuities cause the strong negative

age trend (see Fig. 1). In particular the decrease around 2005 is not evident in the other three reanalyses, and coincides with the onset of assimilating MLS temperatures above about 5 hPa in MERRA–2 (Fujiwara et al., 2021). This change appears to affect the BDC globally and equally strong in both hemispheres (Fig. 1). Hence, this example clearly shows how discontinuities in the time series in each hemisphere cancel out in the hemispheric difference, which is therewith a more robust measure of circulation changes.

Strahan et al. (2020) recently proposed that observed hemispheric asymmetries in transport trends are caused by the inter-action between the annual cycle of the BDC and the QBO, resulting in low-frequency variability with 5- to 7-year period. Here, we find the same low-frequency variations in time series of $N_2O$ and mean age from all four reanalyses (Fig. 3). In addition to this variability the time series suggest a persistent change in the hemispheric transport difference starting around 2004. How long this change will last can only be deduced from future observations. However, the presented analysis of climate

model simulations (CCMI REF–C2) supports the view that at least parts of the BDC change is related to internal variability. The distribution of model simulated hemispheric difference in age trends (averaged over 70–10 hPa, 20–80°) shows a large spread of values for 2005–2017 between about $\pm0.1$ yr/dec (Fig. 10), while the hemispheric difference in reanalysis trends covers the range $0.07 - 0.46$ yr/dec (MERRA–2: 0.07, JRA–55: 0.12, ERA–Interim: 0.31, ERA5: 0.46 yr/dec). Hence, at least qualitatively the observed changes, with an aging NH relative to the SH, are not in contradiction to climate model simulations.

The range of modelled trends depends on the representation of natural climate variability in the models, which might be biased. Hence, a larger number of model ensemble members could likely be necessary to provide the full range of possible changes. However, since the spread in trends between different reanalysis products is large, it is hard to judge whether the observed trend (difference) is outside the range of the climate models in a quantitative sense. In conclusion, the climate model analysis indicates that the hemispherically asymmetric signal in mean age trends in the ozone recovery period is far smaller than the

signal in the ozone depletion period. Therefore the signal is small compared to inter-annual variability for the post-2000 period we have observed to date, and the signal might even be too small to be expected to be ever detected. Hence, the observed hemi-spheric asymmetry in trends is not in contradiction to the opposite expectation from climate models for the ozone recovery period if model variability is taken into account.





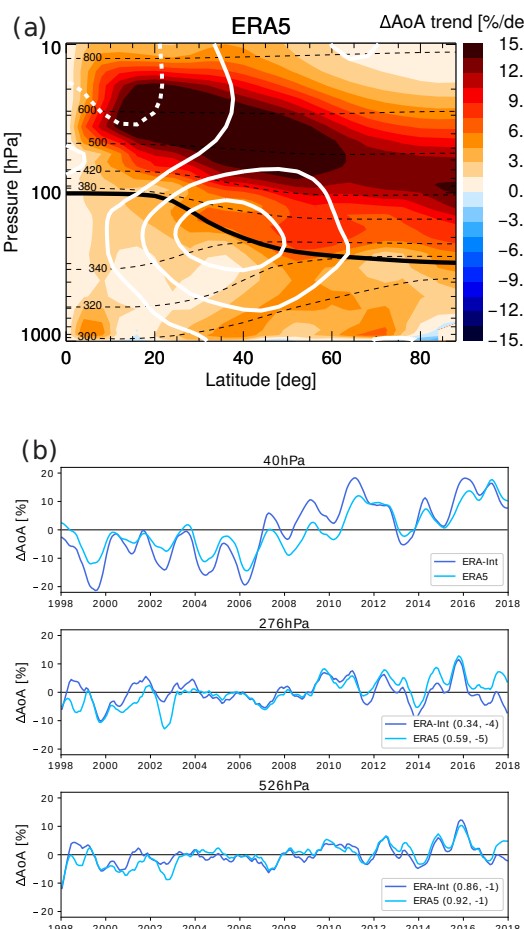

**Figure 11.** (a) Hemispheric difference in mean age trend (NH minus SH) for 2005–2017 as percentage of NH mean age, from CLaMS model simulation driven by ERA5. White lines show the NH zonal wind contours (solid for positive, dashed for negative values). Thin black dashed lines show potential temperature levels and the thick black line the WMO tropopause. (b) Time series of mean age hemispheric difference at about 40 hPa, 276 hPa, 526 hPa, from ERA5 and ERA–Interim simulations. Values in the legend indicate the maximum correlation coefficient between the time series at the given and the next higher level (e.g., in the 526 hPa plot between 276 and 526 hPa) and the corresponding optimal lag (in months).

Remarkably, the hemispheric asymmetry in BDC trends has the potential to influence tropospheric composition by down-

ward transport across the extratropical tropopause. Figure 11 presents the hemispheric difference in mean age trends for ERA5, as percentage change (relative to NH mean age) to better illustrate the downward influence. The figure shows a clear downward propagation of the positive age trend signal from the lower stratosphere into the troposphere, mainly below altitudes of the sub-tropical jets (similar hemispheric asymmetries in tropospheric trends are found for most other reanalyses, not shown). Even at the surface, a positive hemispheric difference in age trends of more than 5% per decade can be found. Time series of mean age





hemispheric difference further show highly correlated variability between the lower stratosphere (here 40 hPa), the lowermost
stratosphere (276 hPa) and the upper troposphere (526 hPa, see Fig. 11b). Correlation coefficients are particularly high between
the tropospheric and lower stratospheric time series and reach 0.92 for ERA5 and 0.86 ERA–Interim between 526 and 276 hPa
(Fig. 11b). A similar influence of stratospheric variability on tropospheric composition has recently been found by Ray et al.
(2019). In addition to the correlated variability we here find a relation of the longer term changes since about year 2000 in the
troposphere and lower stratosphere, with mean age increasing at all levels.

Variations in stratospheric circulation are a major uncertainty for estimates of tropospheric emissions, e.g. for chlorofluoro-
carbon CFC–11 (Montzka et al., 2018; Ray et al., 2019), as changes in transport pathways through stratospheric sink regions
alter the downward flux into the troposphere (e.g., Laube et al., 2020). In particular, the hemispheric difference in tracer con-
centrations, with higher values in the NH than SH, is commonly related to the emissions being higher in the NH than SH. Our
results further suggest that also changes in the hemispheric difference of stratospheric concentrations of long-lived trace gases,
related to hemispheric asymmetries in BDC trends, may propagate downwards and impact the troposphere, likely modifying
hemispheric differences at the surface.

## 7    Conclusions

The main conclusions of the paper are summarized here as answers to the three research questions raised in the introduction:

(i) *How robust are hemispheric asymmetries (differences) in BDC trends over the last about two decades?*

The observed hemispheric asymmetries in BDC trends over the last about two decades (since about 2000), manifesting
in an aging NH relative to the SH, are found to be very robustly represented in current meteorological reanalysis data
sets (here, ERA5, ERA–Interim, JRA–55, MERRA–2). This good agreement in the hemispheric asymmetry of BDC
changes is found although the BDC trends in each hemisphere can largely differ between reanalyses. Furthermore,
the asymmetries in circulation changes cause related asymmetries in trace gas composition changes, and the simulated
hemispheric asymmetry in $N_2O$ trends agrees well with satellite observations.

(ii) *Which processes cause these hemispheric asymmetries?*

The underlying dynamical changes are in good agreement between the various datasets considered. Different residual
circulation and age of air diagnostics indicate a clear structural change in the BDC since about year 2000. This structural
circulation change comprises an upward shifting and strengthening deep circulation branch and a weakening shallow
branch, resulting in a deepening of air parcel's pathways and, subsequently, an aging in the NH relative to the SH.

(iii) *Are the observed recent hemispherically asymmetric BDC changes in contradiction to expectations from ozone recovery
as simulated by climate models?*

The expected long-term response of the stratospheric BDC to ozone recovery, as simulated by free-running chemistry
climate models, would be opposite to observed changes during the last decades, predicting a weakening circulation
and increasing age of air in the SH compared to the NH. However, changes over decadal time scales can be largely



affected by internal variability, and we find that the model simulated range of variability comprises the observed BDC change (aging NH relative to SH). Therefore, the observed trends are not in contradiction to climate model projections, but the recent trends can rather be understood to result from internal variability. As the effect from ozone recovery on hemispheric asymmetries in BDC changes is weak over time scales of a decade and thus small compared to inter-annual and multi-annual variability, this effect can only be expected to be ever detected far in the future (about end of this century).

Our results underline that a correct representation of BDC trends and variability in models is important to place observed composition changes into context of forced trends versus (decadal) variability. This holds for both stratospheric and tropospheric composition, since stratospheric anomalies may impact the troposphere. In particular, hemispheric asymmetries in the BDC and related asymmetries in stratospheric composition, including the observed asymmetries in trends during the last about two decades, are found to have the potential to propagate into the troposphere and affect composition down to the surface.

*Data availability.* ERA5 and ERA-Interim reanalysis data are available from the European Centre for Medium-range Weather Forecasts, JRA–55 reanalysis from the Japanese Meteorological Agency, and MERRA–2 from the National Aeronautics and Space Administration. The CLaMS model data used for this paper may be requested from the corresponding author (f.ploeger@fz-juelich.de). The CCMI–1 data used in this study can be obtained through the British Atmospheric Data Centre (BADC) archive.

*Author contributions.* FP carried out the CLaMS simulations and the respective analysis, HG carried out the analysis of CCMI data. Both authors wrote the paper.

*Competing interests.* The authors declare that they have no conflict of interest

*Acknowledgements.* We thank Paul Konopka for providing CLaMS $N_2O$ simulations. We further thank Paul Konopka, Michelle Santee, Kaley Walker and Rolf Müller for helpful comments on the manuscript. We further thank the ECMWF, JMA and NASA for providing reanalysis data, and the MLS and the ACE-FTS teams for providing $N_2O$ satellite observations. The Atmospheric Chemistry Experiment (ACE), also known as SCISAT, is a Canadian-led mission mainly supported by the Canadian Space Agency and the Natural Sciences and Engineering Research Council of Canada. We acknowledge the modeling groups for making their simulations available for this analysis, the joint WCRP SPARC/IGAC Chemistry-Climate Model Initiative (CCMI) for organizing and coordinating the model data analysis activity, and the British Atmospheric Data Centre (BADC) for collecting and archiving the CCMI model output. This study was funded by the Helmholtz Association under grant no. VH-NG-1128 (Helmholtz Young Investigators Group A–SPECi) and under grant no. VH-NG-1014 (Helmholtz Young Investigators Group MACClim). Finally, we gratefully acknowledge the computing time for the CLaMS simulations which was granted on the supercomputer JURECA at the Jülich Supercomputing Centre (JSC) under the VSR project ID JICG11.



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
