# Peer review of "Hemispheric asymmetries in recent changes of the stratospheric circulation"

_Atmospheric Chemistry and Physics, 2021_

## Author Comment (AC1)

**Reply to Reviewer 1**

We thank the Reviewer for the careful reading and evaluation of the manuscript and the good comments. In the following, we address all comments and questions raised (Reviewer's comments in italics). Text changes in the manuscript are highlighted in color (except minor wording changes).

**General comments:**
*This is a very well written paper. Overall, I find the topic, the scientific work and the presentation excellent, and I have very little to comment. The topic of the paper is timely and contributes directly to current open questions in the field of stratospheric circulation and its impact on (tropospheric) climate change. The scientific approach, with CLaMS calculations based on various reanalyses and compared to observations that are contrasted to free-running chemistry-climate models, is the perfect choice for separating shorter-scale natural variability from long-term climatological trends. The analysis of the data sets is done thoroughly, with well-adjusted scientific methods and on a sound statistical basis. The presentation of the scientific work is very clear, the paper is very well written, and the argumentation is straightforward and easy to follow. I have a few comments only that are on the level of either minor or even technical corrections (I do not separate between them in the list that follows):*
Thanks for this positive evaluation of the manuscript!

**Minor and Technical comments:**

Abstract, line 1-3: *For the first sentence of the abstract, this statement sounds a bit twisted. Maybe it is better to split it into two sentences like: "The effect of ozone recovery on the BDC is expected [from model simulations] to be a deceleration. However, on the contrary, the BDC has been found to weaken .... in observations...." or something similar.*
Thanks for pointing to that. We agree that the formulation was not clear (as similarly stated by Reviewer 2) and changed the sentence to: "The expected effect of ozone recovery on the stratospheric Brewer-Dobson circulation (BDC) is a slow-down, strongest in the Southern hemisphere (SH). In contrast, the BDC has been found to weaken more strongly in the Northern hemisphere (NH) relative to the SH in recent decades, inducing substantial effects on chemical composition." To make the abstract fit into the word limit, we made a few more minor wording changes.

Para from line 34 to 48: *You should clearly state here whether you discuss the direct radiative effect of ODS on climate (since they are potent greenhouse gases), or the "indirect" effect via ozone destruction/recovery. Some wording (e.g. line 40/41: "... Because polar ozone depletion is strongest in the SH, also the ODS effect on the BDC is stronger in the SH than NH, ...") indicates that you are focussing on the indirect effect via ozone depletion/recovery. Some other wording like (line 37/38: "During the period of ozone depletion (before about the year 2000), when ODS concentrations were increasing due to anthropogenic emissions, the ODS increase caused ....") seem to refer to the GHG impact of ODS. It would be clearer if you said here: " ..... the ozone depletion due to the ODS increase caused ....".*
Thanks for pointing to this unclear formulation. The dominant effect is indeed caused by the ozone depletion due to the ODS increase, as stated at the beginning of the paragraph (with reference to Polvani et al., 2018; Abalos et al., 2019). We clarified the wording as suggested by adding "the ozone depletion due to the ODS increase".

line 38: *"... caused a strengthening of the BDC and hence an intensification of the BDC's response to climate change." This sentence is a bit confusing (at least to me): You say the BDC response was the strengthening, and even the strengthening was increasing? (in the sense of a second and not first derivative)?*
This is exactly what we meant here, namely that ozone depletion caused "...an intensification of the BDC's response to climate change", or in other words that the BDC strengthening, which is the circulation response to climate change, becomes even stronger. To be clearer, we added " (i.e., an intensified BDC strengthening)".

Line 54: *"reanalyses" (typo, plural)*
Corrected.

Figure 1, figure caption: *I do not see any "grey lines" in these plots. Please explain what the hatching means.*
Thanks for the careful reading! We removed the grey lines from the plots in the final iteration, to enhance clarity of the first figure, and didn't modify the caption accordingly. The caption has been corrected now, and also the hatching explained: "Regions where trends are not significant are hatched (i.e., where trend values are below 2-$\sigma$, with $\sigma$ being the standard error of the linear trend)."

Line 157: *"In the SH, ERA5 and ERA–Interim show negative trends while JRA–55 shows positive trends ..." This state-*

*ment should be restricted to some areas, e.g. mid-latitudes below  600 K.*
We agree that the description here was unclear. The text has been clarified to: "In the SH, JRA–55 shows positive trends, while the other reanalysis show different trend patterns with regions of both positive and negative trends."

Figure 2, figure caption: *Here I can see the grey lines! Please explain what the hatching means.*
The explanation of the hatching has been added.

Figure 3, figure caption: *The N2O time series seem to be anomalies, given the vmr values ranging between -40 and 40 ppbv. Similar (also anomalies) for AoA?*
Yes, the time series shown in the figure are deseasonalized anomalies. This information has been added to the caption text. Thanks for pointing to that!

Line 216/217: *I am not a native english speaker, however this sentence sounds somewhat weird to me: "indicating a strengthening deep BDC branch above about 10 hPa and a weakening meridional circulation below ....". Do you mean " ... a strengthening OF the deep BDC .... and a weakening OF the meridional ..."*
Corrected.

Line 231/232: *the wording "since about 2000 ..." is a bit sloppy. I think you should be more careful with the exact timing. Strahan et al. (2020) pointed out that the period 2005 to 2012 was a very special one, with an extremely steep "trend"; for the longer run, the NH-SH differences of several quantities show an oscillatory behaviour with an overlaid negative tendency. Your statement should probably stick to the time period actually analyzed (i.e. 2005 to 2017).*
To be clearer here, we modified the sentence to "... the deep branch strengthens over the considered period ...".

Figure 8: *I think this figure needs a bit more explanation. I am aware that one of the authors (FP) has used this way of presentation for the age spectra in previous publications. However, what is shown here is the trend of age spectra, and for some readers who are not familiar with previous papers of the authors, the way of presentation may be a bit hard to digest. Some explanatory lines in the caption or even in the text would be helpful.*
Thanks for noting that. We included more explanation in the figure caption: "Black thin lines show contours of the climatological age spectra, the colour shading shows the trend. Each horizontal cut represents the climatological age spectrum (black lines) and its trend (color shading) at that particular latitude."

Line 282/283: *I would add "negative": "... the mean age trends are generally stronger NEGATIVE in the ozone depletion period (1965–2000) ... ".*
Changed as suggested.

Figure 9, figure caption: *The meaning of the thin black solid and dashed lines needs to be described.*
This information has been added to the caption: "Thin black solid lines show climatological mean age, dashed lines show NH trends."

Line 310: *Is it just questionable, or can it be concluded from this investigation that over the 21 century observations will not provide evidence on a potential difference between NH and SH age trends?*
As the analysis is based on model simulations we think it is better to stay with the more tentative formulation "questionable".

Figure 10: *Explain the meaning of the whiskers.*
The information has been added: "Black stars show the multi-model mean, the boxes indicate the lower to upper quartiles, the whiskers the highest and lowest values, and the orange line the median value of the distribution."

Line 344: *It would probably be helpful to mention the "seemingly discontinuous decreases" earlier, e.g. during discussion of Fig. 3.*
We added a related statement at the end of Sect. 3: "In particular, MERRA–2 shows strong decreases around 1995 and around 2005 in the symmetric contribution but not in the hemidpheric difference."

Line 344: *a bracket "(" is missing before "Fig. 3".*
Corrected.

Line 352: *"Here we find the same low-frequency variations ..." I am not convinced, at least I do not see the same vari-*

*ations as in the Strahan et al. paper. I agree that the dramatic change of N2O between 2005 and 2012 (as seen by Strahan et al.) is also present in the N2O time series here. The other oscillations shown by Strahan et al., however, are much weaker. In AoA the signal is even weaker than in N2O. I think the statement in this sentence should be somewhat mitigated.*

We agree that the formulation regarding the comparison to Strahan et al. was somewhat too strong here. The oscillations in the N2O and age time series shown here are indeed similar to those presented by Strahan et al. for HNO3 and HCl, but the amplitude is generally weaker (in particular for mean age). Therefore, we mitigated the formulation to: "Here, we find similar low-frequency variations in time series of $N_2O$ and mean age from all four reanalyses (Fig. 3), but with a weaker amplitude compared to Strahan et al. (2020)."

*Line 353/354: " ... the time series suggest a persistent change in the hemispheric transport difference starting around 2004....". I think the time series of N2O is a bit over-interpreted. And I don't see this strong change from 2004 on in the time series of AoA in Fig. 3.*

The N2O time series hemispheric difference in Fig. 3c indeed shows a decrease starting in about 2004, and that anomaly values stay largely below the zero line until end of the cosidered period. Therefore, the change in hemispheric difference continues at least until 2020. However, we mitigated the formulation here somewhat by deleting "persistent", and think that the sentence now is adequately describing the results (e.g., Fig. 3c): "time series suggest a change in the hemispheric transport difference starting around 2004.".

*Figure 11, Figure caption: "(b) Time series of mean age hemispheric differences ..." The text says that this is the time series of the hemispheric difference of AoA TRENDS (lines 370, 372, 374), while the figure caption says it is the difference between hemispheric AoA. Please clarify. According to the units it is probably the difference of AoA and not the difference of its trend.*

Fig. 11a shows "Hemispheric difference in mean age trend" while Fig. 11b shows "Time series of mean age hemispheric difference", as explained correctly in the caption. The text parts stated here (lines 370, 372, 374) refer to Fig. 11a and are therefore correct in saying "hemispheric difference in mean age trends". In the following text part, referring to Fig. 11b, it is written "Time series of mean age hemispheric difference", which is also correct. Hence, as far as we can see, the text is correct.

---

## Author Comment (AC2)

**Reply to Reviewer 2**

We thank the Reviewer for the careful reading and evaluation of the manuscript and the good comments which helped to further improve the paper. In the following, we address all comments and questions raised (Reviewer's comments in italics). Text changes in the manuscript are highlighted in color (except minor wording changes).

**General comments:**

*This paper examines changes in the stratospheric circulation via CLaMS simulations forced by different reanalysis products, free running model output from CCMI runs and satellite observations of N2O. This wealth of information is used to elucidate the hemispheric asymmetry in the stratospheric circulation variability over recent decades and put these changes in context of the long-term changes expected due to ozone recovery and climate change. The results are generally consistent with previous studies but this study brings more detail and explanation of the circulation asymmetry than has been done before. The methods of analysis are clearly explained, the figures show the features well and the conclusions are fully justified. I recommend publication of this paper in its current form with consideration of the minor comments listed below.*

Thanks for this positive evaluation of the manuscript!

**Specific comments:**

Lines 1-3: *The first sentence of the abstract is a bit awkward. Perhaps, 'The stratospheric Brewer-Dobson circulation (BDC) has been found to have weakened in the NH relative to the SH in recent decades, despite ozone recovery over this period that would be expected to cause the opposite trend, inducing substantial effects on chemical composition'.*

Thanks for pointing to that. We agree that the formulation was not clear (as similarly stated by Reviewer 1) and changed the sentence to: "The expected effect of ozone recovery on the stratospheric Brewer-Dobson circulation (BDC) is a slow-down, strongest in the Southern hemisphere (SH). In contrast, the BDC has been found to weaken more strongly in the Northern hemisphere (NH) relative to the SH in recent decades, inducing substantial effects on chemical composition." To make the abstract fit into the word limit, we made a few more minor wording changes.

Line 37: *maybe add 'increasing' before 'ozone depletion' here since ozone depletion has been ongoing after 2000. It might be helpful to come up with a term to describe the ozone depletion before 2000 since you refer to it again later.*

We added "increased ozone depletion" for clarification.

Line 63: *I think you meant 'BDC decrease' rather than 'increase'.*

Yes indeed - corrected! Thanks for noticing that!

Line 84: *The 'e.g.' seems oddly placed after the reference.*

Corrected.

Line 157: *It looks like both ERA-Interim and ERA5 have mostly positive trends in the SH.*

We agree that the description here was unclear. The text has been clarified to: "In the SH, JRA–55 shows positive trends, while the other reanalysis show different trend patterns with regions of both positive and negative trends."

Line 162: *'extent' instead of 'extend'*

Corrected.

Line 164: *'of' instead of 'for'*

Corrected.

Lines 216-7: *This sentence could use a bit more explanation. When you say a 'strengthening deep BDC branch' and 'weakening meridional circulation' I assume you're referring to just the NH circulation but it's not entirely clear. And the consistency with the age of air trends is maybe not straightforward since there are positive age trends in ERA5 in the NH at all levels above 100 hPa. Positive age trends (plus negative N2O trends) and a stronger circulation above 10 hPa don't immediately follow so it would helpful to at least make reference to the later discussion in Section 4.2.*

We agree that at this point the relation between residual circulation and age of air trends is not straightforward, and clarified the text with a reference to the later section where the relation becomes clear: "Hence, residual circulation meridional velocity and EP-flux divergence changes are consistent, indicating a strengthening deep

BDC branch above about 10 hPa and a weakening meridional circulation below in the NH relative to the SH. These trends are largely consistent with the trends deduced from age of air and $N_2O$, as will become clear from the discussion in Sect. 4.2."